# A System Dynamics Stability Model for Discrete Production Ramp-Up

Julian Haller [1,*], Bharath Kumar [1], Amon Göppert [1] and Robert H. Schmitt [1,2]

1   Laboratory for Machine Tools and Production Engineering (WZL), RWTH Aachen University, 52074 Aachen, Germany
2   Fraunhofer Institute for Production Technology IPT, 52074 Aachen, Germany
*   Correspondence: julian.haller@wzl-iqs.rwth-aachen.de

**Abstract:** Manufacturing companies are increasingly challenged to deliver customizable products with shorter time to market and higher quality while adhering to sustainability requirements. To meet these challenges, the frequency and importance of production ramp-ups will increase in the future. However, most ramp-ups still fail to meet targets due to unpredictable equipment failures, operator errors, and system complexity. We propose a system dynamics model that captures the unique dynamics of ramp-up phases by integrating stability and disturbance factors that influence the key performance indicators overall equipment effectiveness, process capability, and production output. A systematic literature review informed the identification of stability factors, which were validated through expert interviews in the automotive industry. Our system dynamic simulation results indicate that control factors realistically influence production system behaviour during different ramp-up phases. Despite some limitations regarding the effects of maintenance personnel and engineering changes on key performance indicators, our model effectively simulates realistic ramp-up behaviour. The findings highlight the need for tailored models that consider specific ramp-up contexts and emphasize the importance of data acquisition for enhanced performance prognosis in future research.

**Keywords:** manufacturing; production ramp-up; uncertainty; modelling; system dynamics





## 1. Introduction

Manufacturing companies still face the challenge to deliver products that are increasingly customisable, with a shorter time to market and of higher quality [1,2]. Additionally, sustainability requirements oblige companies to rethink their designs towards a circular economy, more sustainable materials, and more resilient supply chains [3]. To meet these challenges, the frequency and importance of production ramp-up will increase in the future as products are developed and introduced in shorter periods [4,5]. A company that introduces a product to the market at a larger volume and faster than its competitors can accrue greater profits [6]. Conversely, a delayed product launch leads to a loss of profits [7]. Despite this apparent significance of the ramp-up phase, 60% of production ramp-ups fail to meet their quality, time, or cost targets [1,8]. Authors in [9] identified *unpredictable equipment failures*, *increased downtime and subsequent maintenance operations*, *operator errors*, and *events that require restoring to a previous state* as factors contributing to ramp-up failures.

These failures are attributed to the dynamics and complexity of a production system undergoing a ramp-up [10,11]. Consequently, the process of production ramp-up is frequently described as an inherently unstable phenomenon [8,12] that is particularly important for discrete manufacturing, such as in the automotive industry, due to the numerous process steps that must be coordinated [13].

Production systems in the ramp-up phase behave differently from those in stable series production. Authors in [8] characterise ramp-up as a phase with lower knowledge, output, production capacities, and planning reliability and simultaneously higher cycle times,

demand, and disturbances. Coping with the unique issues of the ramp-up phase requires custom-tailored tools and thus, a variety of methods and tools have been proposed over the last two decades: general and risk-based frameworks [14,15], lean [16–18] and agile [19–23] frameworks, cybernetic approaches [24], knowledge and information management [25–29], machine learning [30–33], and simulation [34,35].

Some authors envision a holistic ramp-up assistant, that solves the aforementioned ramp-up challenges through data-based methods in real time [36]. However, ref. [1] concluded that the lack of data during the ramp-up phase is still not solved. In [37], we further inferred that the lack of common data models for the ramp-up phase inhibits the deployment of digital twins and sufficient data acquisition. Thus, we still judge the use of models and simulations essential to ramp-up planning, control, and generation of synthetic data as the backbone for data-based methods.

In [38], we evaluated suitable methods for modelling production ramp-up for preventive or reactive actions. Researchers have employed various methods, such as mathematical models, discrete event simulations, or Markov models. However, many models are limited to isolated ramp-up issues, for example capacity or personnel planning, and thus do not reflect ramp-up complexity [12]. The system dynamics modelling approach was designed specifically with system complexity in mind [39] and thus we propose a system dynamics model for production ramp-up to complement other modelling approaches.

This paper is structured as follows: Section 2 covers related research, Section 3 describes the proposed methodology for a system dynamics model, Section 4 presents the results of the system dynamics simulation and validates results on an industry use case, and Section 5 discusses the results and concludes with future research opportunities.

## 2. Related Research

We performed a systematic literature review to identify stability and simulation models for production ramp-up. The review was executed according to the PRISMA methodology [40] in the databases *Scopus* and *Web of Science.* We screened title, abstract, and keywords with the following search string:

$$\text{(``ramp-up'' OR ``start-up'') AND (stabil* OR disturb* OR disrupt*) AND} \tag{1}$$
$$\text{(manufact* OR assembl* OR produc*)}$$

Peer-reviewed articles in English after the year 2000 were eligible.

The stability of production ramp-ups remains vague regarding how it is determined. Several researchers tried defining it with varying granularity. On a strategic level, ref. [41] propose, that lower "product complexity and newness [. . .] and higher levels of maturity are associated with better ramp-up performance". Ref. [42] disproved their hypothesis that product newness leads to better ramp-up performance.

Refs. [11,43] assess ramp-up stability from a complexity perspective. In [11], factors from *product*, *process*, *network*, *organisation*, and *people* domains are rated from low to high complexity to determine stability before ramp-up. The production system then ramps-up in discrete steps to mitigate complexity and guarantee stability. Ref. [43] rate only six factors' complexity: *scope of change and product maturity*, *process complexity*, *affected stations*, *linking and layout change*, *IT system and control technology change*, and *supplier experience.*

On a more tactical level, researchers determine stability through fulfilling KPIs. Ref. [44] derive ramp-up stability from process capability and develop a framework to select production technologies that are most likely to be capable. Ref. [45] develop a multivariate capability index, since traditional CPIs cannot display interdependent cause-and-effect relationships.

On the most operational level, researchers provide a comprehensive overview of stability and disturbance factors. Ref. [46] concentrate on production system design and list *technical competency*, *supplier relationship*, *product and process expertise*, and *organisation improvement culture*, *etc.*, as stability factors. Ref. [47] list 45 key influencing factors within *network*, *location*, *process*, and *product* and rate their complexity, similarly to [11,43]. They

further list disturbances of domains *human*, *machine*, *material*, *process*, and *environment* and associated preventive and reactive measures. Ref. [43] list ten disturbance factors, that affect OEE.

Ref. [12] conducted a systematic review of quantitative decision support during production ramp-up, among them simulations about stability aspects. For instance, capacity planning [48,49], worker assignment [34], workflow management [6,50], or performance measurement [51]. Ref. [12] stated the modelling of several ramp-up objectives as a major research gap.

Only a few simulation approaches considered system dynamics to model production ramp-up. Ref. [52] built a model on a strategic level, that puts variables such as *product innovation*, *demand uncertainty*, *market competition*, and *customer desire* into perspective. Ref. [53] model causal relations and stock flows of company knowledge management during new product introductions. The model considers variables such as *knowledge sharing degree*, *transfer mechanism*, *cultural distance*, and *company innovation desire*. Ref. [54] developed a basic operational SD model to establish variable relations between research and development and production departments. *Yield*, *capability*, *engineering changes*, *cycle time*, and *throughput* are part of their KPI system, but the authors conclude that it is only a basic model that ought to be expanded. Ref. [55] use SD to model the assembly cost for reconfigurable automated or manual assembly systems. Overall, these developed SD models consider only isolated ramp-up objectives or remain very simple. To the best of our knowledge, ref. [56] was the first to develop a coherent SD model that combines operational objectives with strategic ones, such as learning behaviour. Ref. [57] built his SD model on top of [56]. Both authors did not disclose the underlying model equations.

Consequently, the model we propose is the most comprehensive SD model regarding production ramp-up to date. It is the first SD model to be built upon a systematic literature review and to be validated by expert interviews and industry data.

## 3. Materials and Methods

### 3.1. Descriptive Stability Model

The goal is to simulate the stability behaviour of ramp-up systems. There are many different perspectives on what is considered stability during ramp-up, as we pointed out in [58]. In this work, we consider four key performance indicators (KPI) to assess ramp-up stability: *overall equipment effectiveness* (*OEE*), *production process stability*, *time-to-volume*, and absolute *production output*. First, we identified the necessary components for the system dynamics model that influence these KPIs, hereinafter referred to as modules. The modules are categorized according to the 5-M: *man*, *machine*, *material*, *method*, and *milieu*. Stability factors within their 5-M category are depicted in Figure 1. The factors are based on our systematic literature review and validated in the results section.

Within the category of *man*, we evaluated *relative work intensity* [59], *maintenance personnel* [60], *production personnel* [61], and *worker skill* [50,60,62] as the most important factors for stability. *Machine breakdowns* are the major disruption cause during ramp-up [63] and reducing breakdowns has a direct impact on ramp-up performance [12]. We focused the *material* category on the product, thus comprising *product defects* [60] and *product maturity* [64]. Both are intertwined: as the maturity decreases, product defects will increase [65]. Furthermore, insufficient specifications of recent product development contribute to an increase in defects [66]. Following design changes cause further performance losses [60]. Moreover, we included *process maturity* as the sole factor of the category *method*. Low process maturity leads to more instabilities [67] and leads to more defects [63,66] and machine breakdowns [60]. Lastly, the *milieu* category comprises *production demand*, *engineering changes*, and *complexity drivers*. *Production demand* is defined as the internal orders for the production system [68] and thus influences several disturbance factors [60], such as relative work intensity or machine breakdowns. Higher production demand further requires a sufficient process maturity [69]. *Engineering changes* are a consequence of insufficient product or process maturity [70]. Late changes result in lower production output and

delayed deliveries and thus ramp-up instability [64]. *Complexity drivers* map ten structural complexity drivers such as *product novelty* or *assembly system size*, according to [65], that are inherent to the specific ramp-up instance.

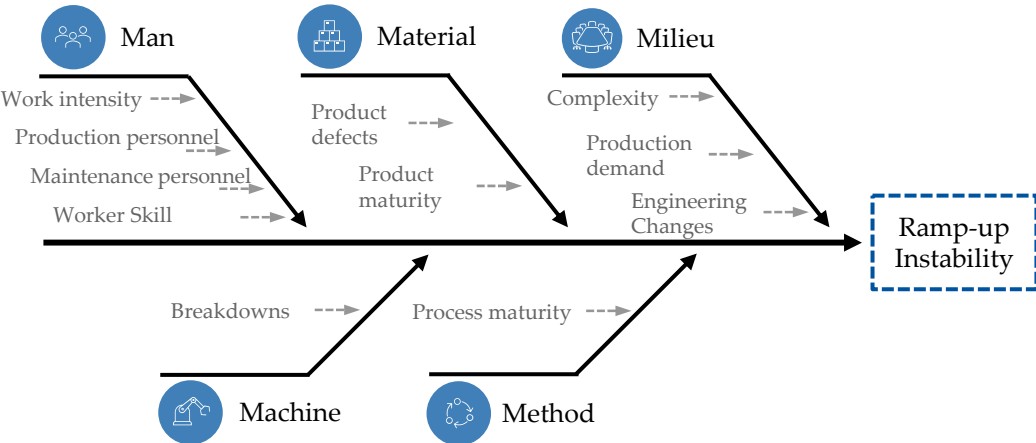

**Figure 1.** Ishikawa diagram of (in-)stability factors during production ramp-up. Eleven influencing factors were identified in a systematic literature review and categorized according to 5-M.

We selected four major control factors to influence the system, based on the current instability state. Firstly, both the number of production and maintenance personnel (and thus the ratio between them) can be adjusted. This is analogue to [57], who divided control factors into *value-adding* and *improvement* measures to reflect the goal conflicts during ramp-up. Furthermore, we defined *engineering changes* and *training time* as control factors. Engineering changes might be required to react to unforeseen instabilities due to product design flaws or suboptimal process parameters. While production output is lower for a while, the ramp-up system can be more stable in the long term when changes are made. Similarly, training blocks operators from production or maintenance but increases worker skill that counteracts instability.

*3.2. System Dynamics Simulation*

As we concluded from our systematic literature review, addressing the multitude of production ramp-up instability factors remains an unanswered research question. We judge system dynamics as the most suitable method for simulating the multitude of factors. Furthermore, system dynamics models attach particular importance to information flows, allow the integration of fuzzy aspects of the system, and can handle nonlinear processes well. Therefore, we favour system dynamics over discrete event simulation, which is well suited for detailed analysis of processes that are characterized well and leave less room for uncertainties.

Jay W. Forrester introduced system dynamics in 1961 to model complex social organizational systems [39]. System dynamics models have two typical representations: causal loop diagrams (CLDs) and stock-flow diagrams (SFDs). Figure 2 shows elements of both representations.

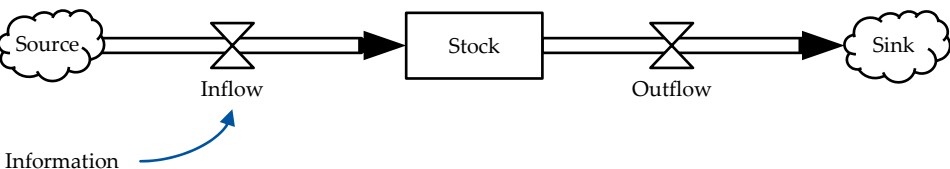

**Figure 2.** Representation of system dynamics basic elements. The blue arrow indicates that the variables *information* and *inflow* have a causal relationship. Variables *inflow* and *outflow* determine how much value of *source* flows into *stock* and out of *stock* into *sink* over time.

CLDs consist of cause variables (indicated by the information variable) and effect arrows (indicated by blue arrow). Effect arrows are directed in one direction, i.e., an arrow can indicate only one effect between two variables. The effect of one variable can either be positive or negative to another variable. In the case of Figure 2, the information variable affects the inflow variable.

SFDs consists of stocks, represented as a rectangular box, and flows that are indicated by valves in Figure 2. The structure of SFDs is analogue to fluid mechanical systems, with flows symbolizing pipelines that facilitate the flow of fluids between stocks. The direction of the flow is indicated by the flow arrow. Stocks are represented by containers storing fluids and serve as variables that reflect the accumulation of a flow variable over time. The value of flows indicates the change in the accumulation of stocks for future time periods. Lastly, a system dynamics model includes sources and sinks. These are special types of stock variables outside of the system boundary. For a deeper understanding of system dynamics, we refer the reader to [39] or [71].

Following our descriptive stability model and the system dynamics methodology, we developed eight new modules, contemplating the existing work of [56,57], resulting in eleven total modules. The system dynamics model was developed with *Vensim PLE 10.1.3* by *Ventana Systems Inc.*, Harvard, MA, USA. Figure 3 provides an overview of all modules and the connections between them.

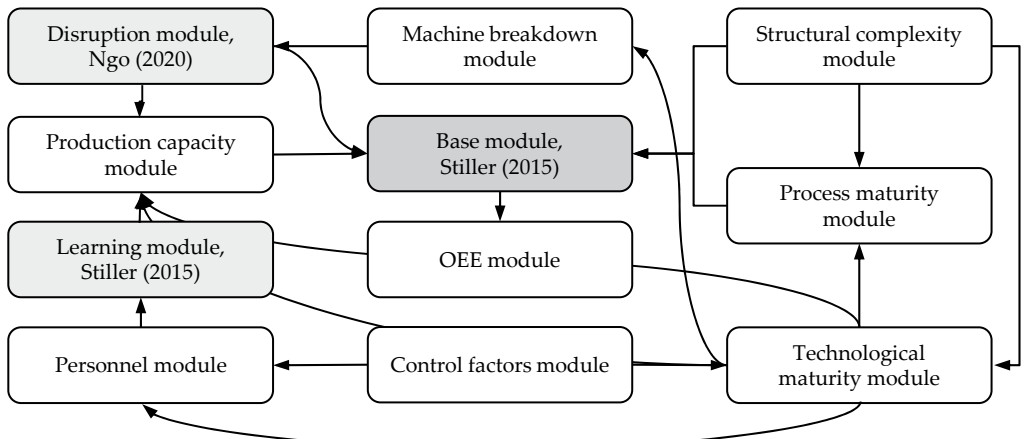

**Figure 3.** Schema of all modules of the system dynamics model. The arrows indicate what modules are connected through common variables. The base and learning modules were imported from [56] and the disruption module was imported from [57].

The imported modules were imported logically and remained unchanged. However, the newly developed modules introduced new interactions and dependencies. New modules were first developed separately based on literature and expert input. The new modules were subjected to a rigour cycle, i.e., they were tested for internal consistency before integration and parameters were adjusted after integration to comply with the overall SD model.

### 3.2.1. Base Module

In the following sections, we present every module in detail. All variable equations are provided in Supplementary Material S1. The work of [56] developed a quality-oriented ramp-up model, which we condensed into our base module that is shown in Figure 4. It describes a basic material flow from a *planned production* stock to a *finished goods* stock via a *work-in-progress* (*WIP*) stock. From the WIP stock, the *failure rate* and *failure detection rate* determine how many parts flow into the *defective parts* stock. Planned production is affected by demand and there is a possibility that finished goods must be modified. For an in-depth explanation of the module, we refer to [56]. Variables that are shown in

brackets < > originate from another module but affect a variable of the current module and thus form the connections between different modules throughout the SD model.

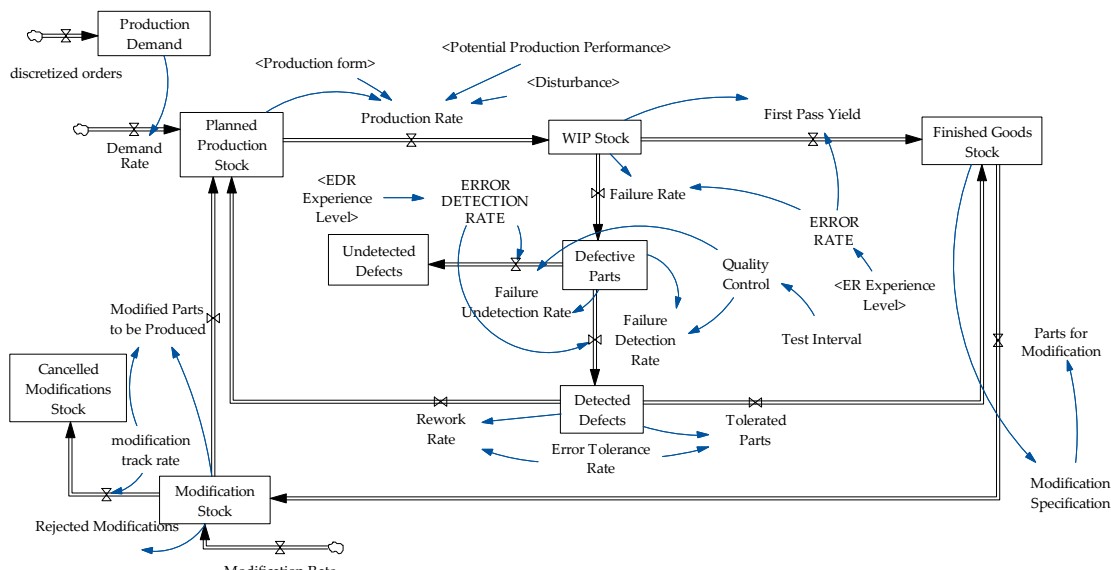

**Figure 4.** Base module of the system dynamics model. It determines how many parts are to be produced at what rate, how many defects occur and how many are detected, and how many defective parts are reworked or modified to eventually qualify as finished goods.

The production system reacts to a pull-based order system, which is modelled by a demand rate variable [68] that we added to Stiller's base model. The demand rate is modelled according to a traditional ramp-up sigmoid curve and follows Equation (A1) in Appendix A.

### 3.2.2. Learning Module

We further imported the learning module of [56], which is depicted in Figure 5. It captures the increase in worker experience over the ramp-up period. The learning experiences are categorized into three different mechanisms: learning from the assembly system's *error rate* (ER), learning from the *error detection rate* (EDR), and learning from the sole production of parts (*production rate* (PR)).

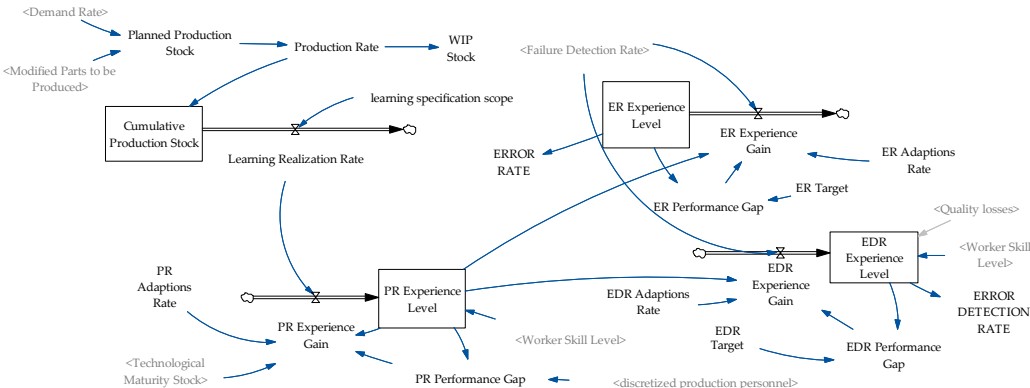

**Figure 5.** Learning module of the system dynamics model. It determines how production workers gain experience from producing parts and detecting errors.

The *PR experience level* is implemented based on the *cumulative production* throughout the ramp-up. The *PR experience gain* flow variable is influenced by the *learning realization rate*, the gap in the ideal production experience level of the worker (*PR performance gap*),

and the adaption of learning variables (*PR adaptions rate*). The *PR performance gap* variable is influenced by the number of *production personnel* present in the assembly system. The *PR experience gain* flows into the *PR experience level* stock variable, which represents the cumulative *PR experience gain*. The ER and EDR experience modules are implemented similarly to the PR experience module. The process experience module ensures the integration of experiential learning mechanisms within the ramp-up system.

### 3.2.3. Disruptions Module

The *disruptions* module by [57] further extends the model's capabilities and is illustrated in Figure 6.

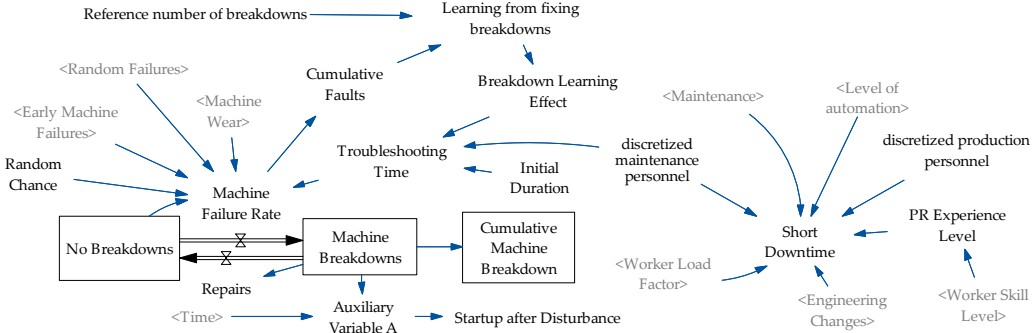

**Figure 6.** Disruptions module of the system dynamics model. It determines what variables lead to machine breakdowns, how long it takes it to resolve a breakdown, and what causes *short downtimes* in production.

This module is implemented to model disruptions of the ramp-up system. The incorporation of breakdowns into the simulation model is achieved through a statistical approach, involving a *random chance* variable triggering machine breakdown. The *machine failure rate* is influenced by the *troubleshooting time*, and if the *random chance* has a higher value than the *troubleshooting time*, a machine transitions from the *no breakdown* state to the *breakdown* state. We further added a Weibull distribution to the *machine failure rate* to account for early, random, and wear failures of machines. The *repairs* flow variable moves machines to the *no breakdown* state, contributing to the overall system stability. *Short downtime* is linked to the number of production and maintenance personnel. This variable accounts for the brief periods of downtime that are quickly resolved, not leading to complete machine breakdowns. Lastly, workers also accumulate knowledge about the system after fixing breakdowns, resulting in a *breakdown learning* effect. For the base functionality of the module, we refer to [57]. We added the capability to model short downtimes due to user errors or engineering change implementation, which affect availability but are not considered a breakdown.

### 3.2.4. Control Factors

To enable interaction with the system, we implemented a designated control factor module. The behaviour of control factors is further discretized into three distinct ramp-up periods, following the definitions of [51,72]. Consequently, the weightage (i.e., priority) of factors adjusts over time. The overall definition of discretized control factors follows that of Equation (A2) in Appendix A, while Equation (A3) in Appendix A provides the example for the discretized production personnel.

During the product-quality phase, the focus is on providing the workers with sufficient training and conducting any engineering changes required to improve the production yield of the ramp-up system. Hence, the weightage assigned to average training time and frequency of engineering changes variables is equal to one. In the production-output-focused phase, the priority shifts to increasing the production output of the system and increasing the number of production personnel. Hence, the weightage of production

personnel is set to one and the weightage of expected production demand is set at 0.8. In the final organizational improvement phase, the emphasis is shifted to the reduction in engineering changes and training time, while maintaining the production yield and output of the ramp-up system. In this phase, the weightage of average training time and frequency of engineering changes are set to 0.1 while the weightage of other variables is set to 1. Table 1 provides an overview of the weightages of the control factors in each phase.

**Table 1.** Weightage of control and input factors according to ramp-up phases.

| Control Factors | Quality Phase | Production Output Phase | Organization Phase |
| --- | --- | --- | --- |
| Avg. Training Time | 1 | 0.33 | 0.1 |
| Engineering Changes | 1 | 0.33 | 0.1 |
| Production Personnel | 0.75 | 1 | 1 |
| Maintenance Personnel | 0.5 | 0.5 | 1 |
| Orders | 0.4 | 0.8 | 1 |

### 3.2.5. OEE Module

We implemented an OEE module to assess system stability as we formulated OEE as one of our main KPI. OEE is extensively defined in [73], and our module implementation is illustrated in Figure 7.

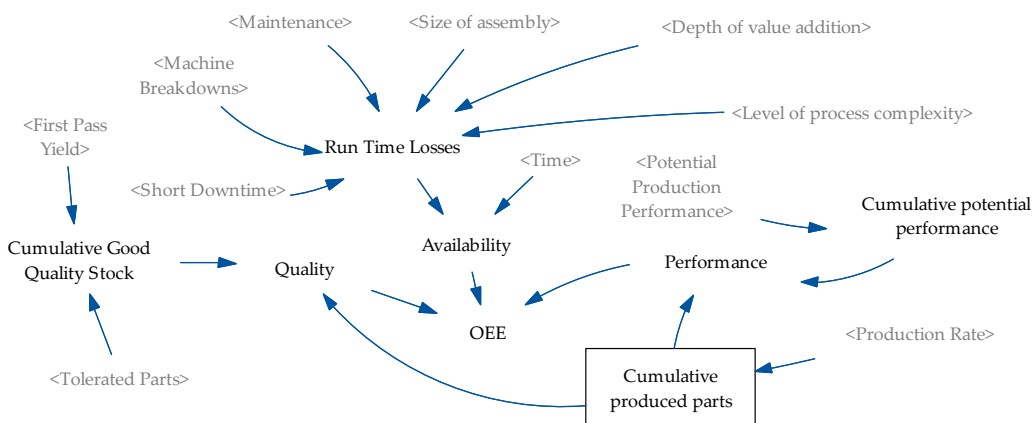

**Figure 7.** OEE module of the system dynamics model. It implements the standard definition of the KPI OEE, consisting of *quality*, *availability*, and *performance*.

### 3.2.6. Breakdown Module

The breakdown module simulates breakdown types along with the variables that influence breakdowns and is modelled as shown in Figure 8. Consequently, the module has a direct effect on the disruptions module. While the disruptions module simulates overall effects on the production system, the breakdown module models what leads to disruptions on a more granular level.

Machine breakdowns can be divided into early failures, wear failures, and random failures, and thus be modelled by a Weibull distribution [63]. All three failure classes directly influence the *machine failure rate. Early machine failures* are affected by *technological maturity*, *worker load factor*, and *Weibull breakdown rate* which depends on passed time of the ramp-up. *Wear failures* are represented by a stock variable *machine wear* that accumulates by the *wear rate* and dissipates by *maintenance* and the number of *maintenance personnel*. The wear rate is influenced by variables such as *worker skill* and the *nominal wear rate* accounting for the number of produced goods.

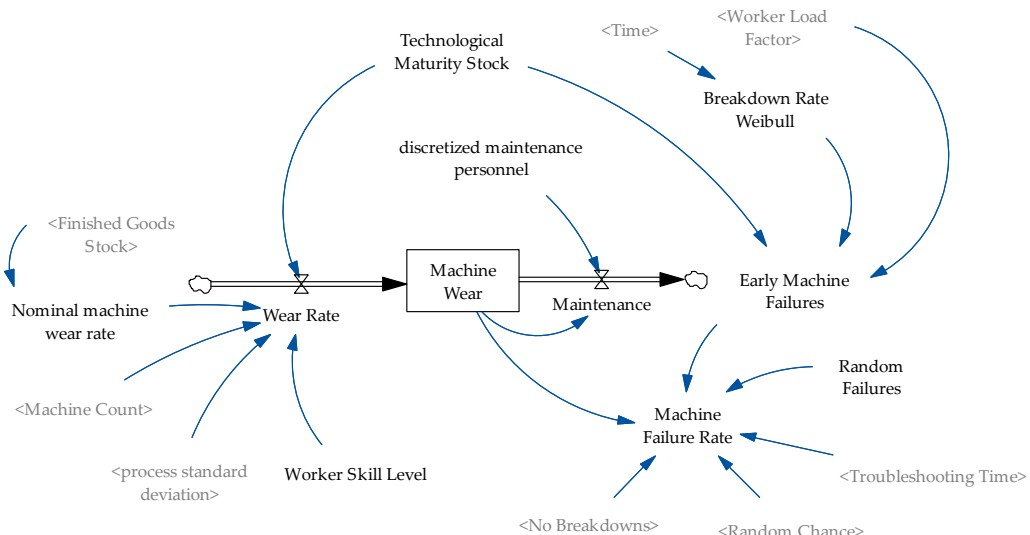

**Figure 8.** Breakdown module of the system dynamics model. It implements a breakdown behaviour according to the Weibull distribution that comprises early failures, random failures, and wear failures. Further, the module models variables that affect the *wear rate* and *machine wear* stock.

### 3.2.7. Technological Maturity Module

This module is implemented to simulate the growing maturity of the product and its effect on the production system and is shown in Figure 9.

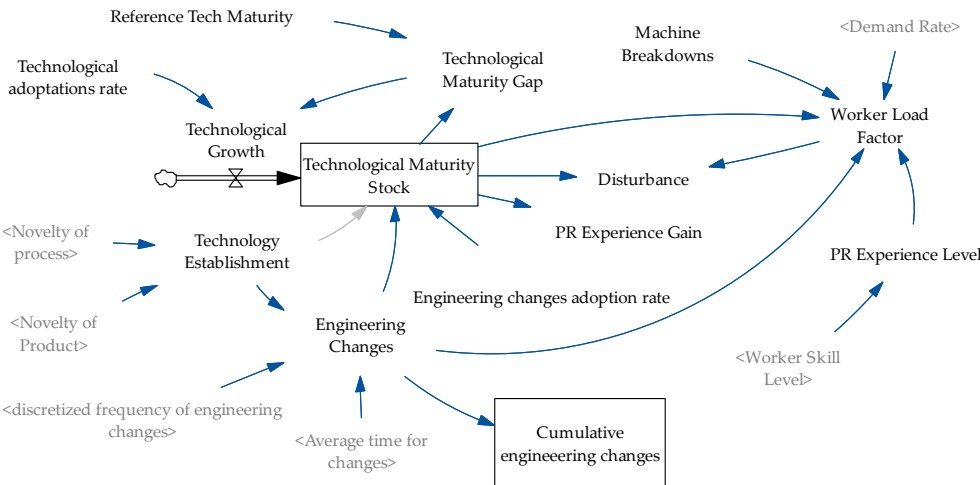

**Figure 9.** Technological maturity module of the system dynamics model. Engineering changes foster technological maturity and thus stabilize production in the long term, but add strain to workers, who must implement changes parallel to running production.

*Technological maturity* stock increases with *engineering changes* in the ramp-up system that influence product and process specifications [64]. While engineering changes lead to a short downtime, the production systems will obtain a higher stability due to higher maturity [61]. Technological maturity is further influenced by *product* and *process novelty* and by *adoption rates* of new technology or engineering changes. Technological maturity affects variables such as process-related *disturbances*, *experience gain* of workers, or the *worker load factor*.

### 3.2.8. System Capacity Module

The system capacity module simulates the potential production performance without any disturbances and is influenced mainly by production capacity and personnel capacity,

whereas the potential production performance is given by the minimum of the two former variables. The module is illustrated in Figure 10.

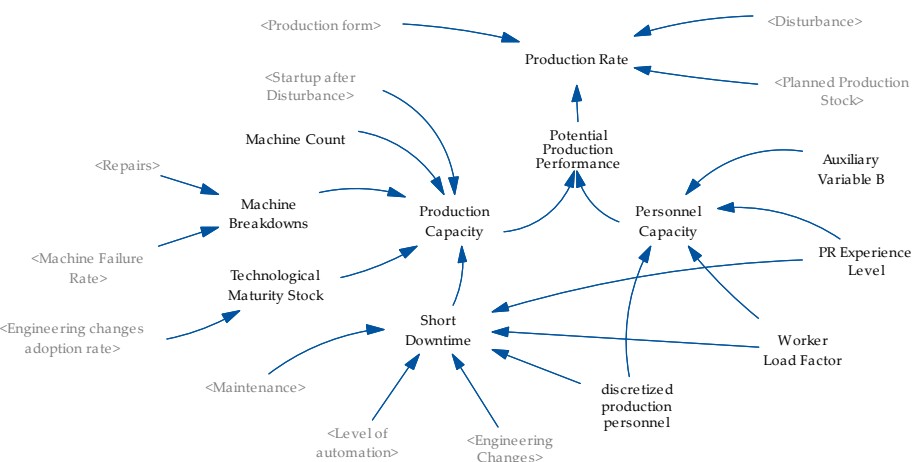

**Figure 10.** System capacity module of the system dynamics model. It determines the theoretical maximum performance of the system based on personnel and technology.

The production capacity variable is influenced by the available machine count, machine breakdowns, technological maturity of the process, and short downtime losses. Personnel capacity is determined by the number of production personnel, the worker load factor, and the process-related experience level.

### 3.2.9. Process Maturity Module

The process maturity module implements the process capability index as part of the model's KPI. It is defined by the specification limits, process performance, and process standard deviation, as illustrated in Figure 11.

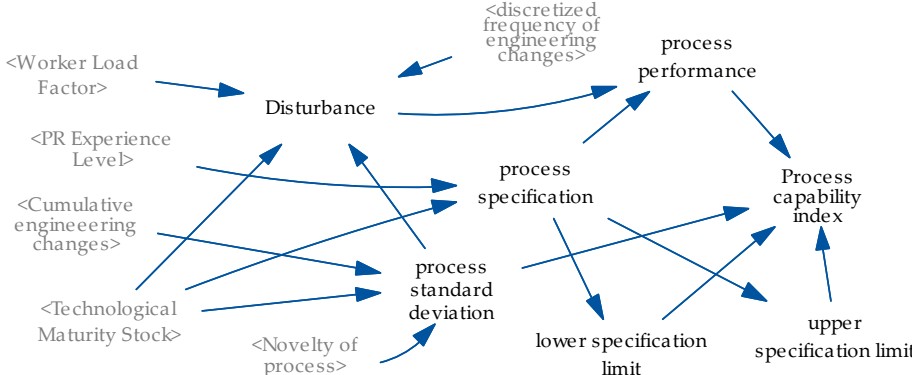

**Figure 11.** Process maturity module of the system dynamics model. It determines the theoretical maximum performance of the system based on personnel and technology.

The *process standard deviation* is influenced by *process novelty*, *technological maturity*, and *cumulative engineering changes*. *Specification limits* are affected by *technical maturity* and *process-related experience*. Lastly, the *disturbance* variable simulates a *process performance* deviation from the specifications.

### 3.2.10. Worker Skill Module

The *worker skill module* models the effect and development of *workers' skill* due to training and learning effects during the ramp-up phase and is shown in Figure 12.

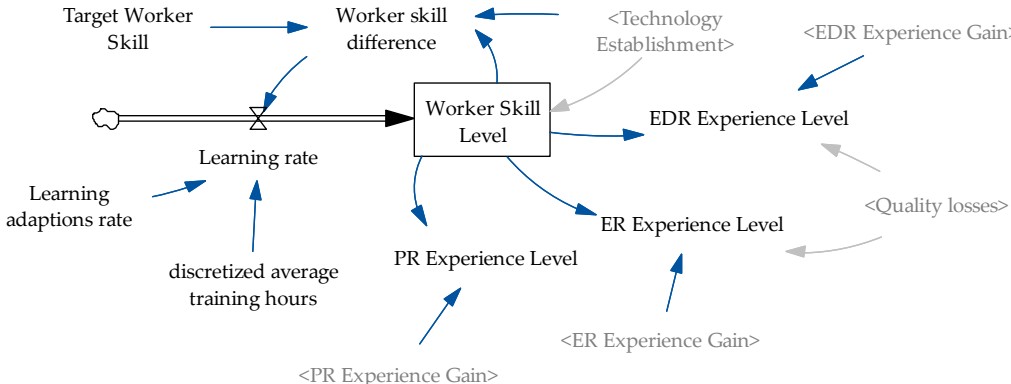

**Figure 12.** Worker skill module of the system dynamics model. It models at what rate workers increase their skill. The learning rate depends on the received training hours and current gap to a desired skill level. Worker skill primarily influences process-related, error rate, and error detection rate experience levels.

The *learning rate* determines how fast workers acquire skill and is influenced by the *average training hours* undergone by the workers. Further, the *learning rate* is bigger when a greater *skill difference* is detected in the beginning of a ramp-up and decays over time as workers get more familiar with product and process [74]. The *worker skill level* influences the experience level of workers regarding *process*, *error rate*, and *error detection rate*.

### 3.2.11. Structural Complexity Module

The *structural complexity module* addresses structural complexity inherent in production ramp-up according to [65]. The ten variables take values from zero to one, depending on the characteristics. Therefore, it acts as a module that takes input parameters and defines different starting conditions for different ramp-up scenarios. The module is illustrated in Figure 13.

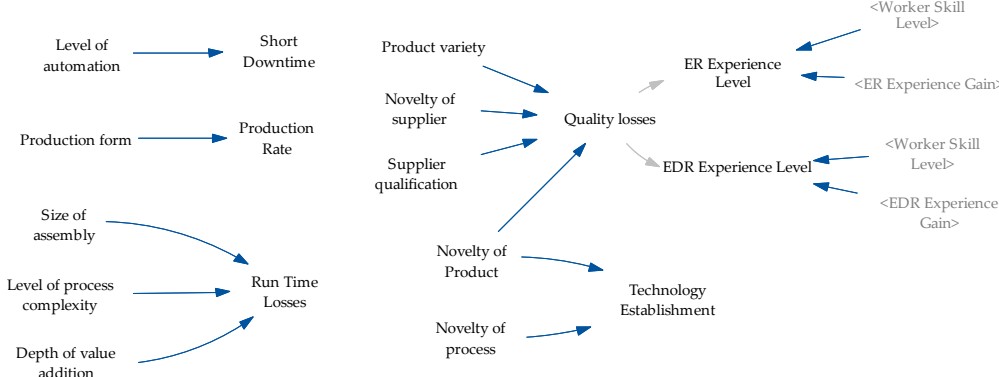

**Figure 13.** Structural complexity module of the system dynamics model. It can be regarded as the starting values of the ramp-up environment, as all ten factors are known before the start of production.

Gartzen's complexity drivers affect five variables of the system dynamics model, which are calculated as averages of their inputs. For detailed definitions, we refer to [65].

## 4. Results and Model Validity

### 4.1. Stability Model Validation

Firstly, we conducted semi-structured expert interviews in the automotive industry to validate if our literature-based stability factors (see Figure 1) are sufficiently relevant in practical use cases. The consolidated results are shown in the Ishikawa diagram in Figure 14.

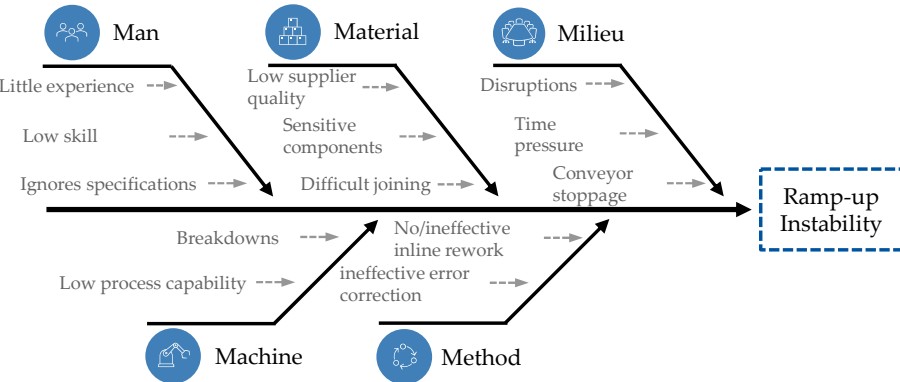

**Figure 14.** Ishikawa diagram of factors that lead to instabilities during production ramp-up, according to automotive industry ramp-up experts. The results mostly confirm our literature-based factors.

Industry experts judge *low worker skill*, *little experience*, *disruptions*, *machine breakdowns*, *low process capability* as instability factors, which are included in our model as well. Some factors are not congruent with our model but can be considered synonymous. For instance, time pressure has similar characteristics to our factor *worker load*. *Difficult joining* and *sensitive components* could be summarized into *low product maturity* of our model. While *low supplier quality* is not explicitly mentioned in our stability model, *supplier qualification* is part of the *structural complexity module*. Similarly, *ineffective inline rework* and *ineffective error correction* are implemented as *rework rate* and *error detection rate* but are possibly not detailed enough for all cases. Solely, the fact that workers might *ignore specifications* or work instructions is not mapped in our model. Therefore, we judge our stability model as sufficiently valid for real industry cases.

### 4.2. Simulation Model Validation

While the simulation model can be adjusted to preferences, we developed it with default values based on automotive industry cases. Therefore, the standard duration of the model comprises a ramp-up period of 150 days with a time step of 6 h. The three ramp-up phases are thus equally divided into 50 days each. We conducted a sensitivity analysis to validate model performance regarding changes in control factors. The control factor changes for the analysis are provided in Table A1 in Appendix A.

The results of the sensitivity analysis show the behaviour of the KPIs *production rate*, *OEE*, and *process capability index* depending on the four control factors *production personnel*, *maintenance personnel*, *average training time*, and *frequency of engineering changes*, as illustrated in Figure 15.

Figure 15a–c show the influence of the control factor *production personnel*. It can be observed in Figure 15a that production personnel influence is strongest on the KPI production rate. The relative impact is greatest in the early ramp-up phase with a 32.8% higher production rate with high settings, and 25.5% lower production rate with low settings, compared to the base run. The impact degrades over time, as the production rate is only 11.8% higher and 18.3% lower in the final ramp-up phase in the respective settings. This is possibly due to increasing *process* and *technological maturity* as well as *worker skill*, so that the sole impact of personnel becomes less defining in the late ramp-up phase.

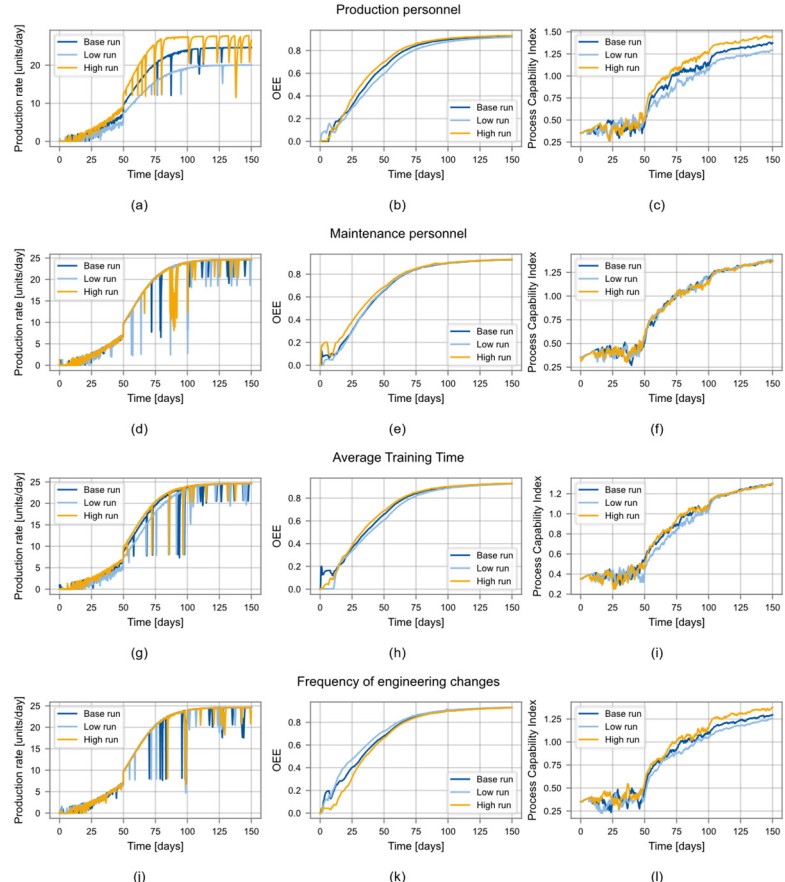

**Figure 15.** Results of the sensitivity analysis regarding the four control factors (rows) and their impact on the three key performance indicators (columns). The first row shows production personnel impact on production rate (**a**), OEE (**b**), and process capability index (**c**). The second row shows maintenance personnel impact on production rate (**d**), OEE (**e**), and process capability index (**f**). The third row shows the impact of average training time on production rate (**g**), OEE (**h**), and process capability index (**i**). The last row shows the impact of engineering changes on production rate (**j**), OEE (**k**), and process capability index (**l**). The analysis shows a large influence of production personnel and training time of the production rate, and high impact of engineering changes on OEE and PCI. Overall, maintenance personnel have only a marginal influence on the system.

Figure 15b illustrates the impact on the KPI OEE. More production personnel have a positive impact on OEE, especially in the first and middle phase of the ramp-up. More available workers reduce the worker load and therefore reduce error probability. However, with maturity advancing, the impact becomes negligible in the late ramp-up phase.

Figure 15c shows the impact on the KPI *process capability*. More production personnel do not have a measurable impact on process capability in the first ramp-up phase, but between +/− 7–10% in phase two and three. This can be explained by the higher system capacity and lower worker load, as explained in the respective modules in the methodology section. Since the *process standard deviation* is influenced by factors such as *technological maturity* as well, the impact of personnel is marginal.

Figure 15d–f show the influence of the control factor *maintenance personnel*. Overall, the impact of maintenance personnel is marginal to the system in its current implementation. Only OEE is significantly affected by the amount of maintenance personnel in the first ramp-up phase, due to the repair of breakdowns. However, breakdowns occur less frequently in phase two and three of a ramp-up due to growing technological maturity; hence, the impact of maintenance personnel reduces.

Figure 15g–i show the influence of the control factor *average training time*. The average training time directly affects *worker skill* and thus variables that profit off worker skill. This effect can be especially observed in the first two ramp-up phases with KPIs OEE and production rate. Worker skill leads to a higher *process-related experience* and *error detection rate* and thus higher *process performance*, which positively affects OEE and production rate. It further positively affects *process capability*, but the effect is only significant in phase two of the ramp-up. Potentially, technological factors have a stronger impact on process capability in the first phase, compared to worker skill improvements.

Lastly, Figure 15j–l show the influence of the control factor *engineering changes*. Engineering changes have a positive impact on process capability throughout all phases of the ramp-up, since they advance *technological* and *process maturity*. The control factor adds up to a 17% gain in process capability in phase one but diminishes to 5% in phase two and three. Since the production needs to stop to implement engineering changes, they have an adverse effect on KPIs production rate and OEE. While production rate is only affected $+/-5\%$ in phase one and neglectable in phases two and three, OEE drops on average 17% in phase one with a high number of engineering changes.

Interpreting the results of the simulation study, there are some limitations regarding the effect of control factors on system behaviour. The effect of maintenance personnel is weak on all KPIs. Furthermore, an extreme test (setting maintenance personnel to 0) revealed that more breakdowns occur (which set reduce production rate to 0 for a while) and the troubleshooting time increases. However, in the case of no maintenance personnel, all machines will breakdown at one point and not recover so that production should come to a complete halt eventually. That event is unlikely in the ramp-up timespan of 150 days but is a shortcoming of the current model implementation.

Furthermore, some control factor effects vary between phases without clear explanation of the causes. For instance, higher *average training*, and thus *worker skill*, *strongly* affect production rate and OEE in the first two phases, but process capability only in the second phase. It can be argued that this is due to stronger influences of technological maturity. There are some issues with the relationship between engineering changes and OEE; the KPI reduces under a high amount of engineering changes in the first phase which is due to stoppages in the production line. However, a high number of changes should result in a higher technological maturity compared to fewer changes and an increase in OEE in the later phases of ramp-up. This is not the case in the current model implementation.

Lastly, we gathered data from an industry ramp-up case to validate the model. Due to confidentiality concerns, we can only provide limited information on the case. The case is a new product *iteration* that is introduced into an existing assembly line with several other variants, where the previous iteration was assembled as well. Production numbers of the new iteration were provided on a weekly basis for the time that was considered the ramp-up and are shown in Figure 16.

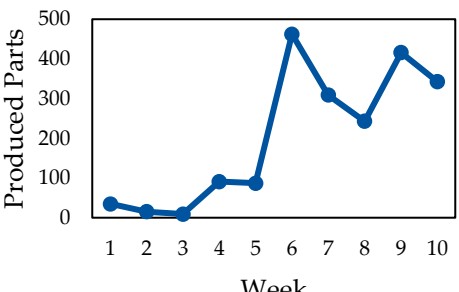

**Figure 16.** Production data of an automotive industry ramp-up case. It shows the number of produced goods over a period of ten weeks. The ramp-up starts slow but gathers momentum after around five weeks. However, disruptions still occur in the final ramp-up phase.

The considered ramp-up period amounts to ten weeks, i.e., 50 working days and thus only a third of our proposed model timeframe. The ramp-up can still be divided into three phases: quality-oriented production in the first three weeks, with 9 to 35 produced units per week; increasing production volume in weeks four and five, with ca. 100 units per week; and full-scale production in weeks six to ten, with up to 500 units per week. However, produced units are significantly lower in weeks seven and eight due to disturbances. Although the used case's environment is different from our model starting point, we infer that our model can model real ramp-up behaviour.

## 5. Discussion

We inferred that there are a lack of simulation tools to model a production ramp-up that address overall ramp-up complexity and the interdependence of instability factors. We identified system dynamics as a suitable tool to address that issue, due to the ability to incorporate fuzzy relationships and provide insights on a strategic level. We gathered instability factors and developed a descriptive ramp-up stability model that was validated with industry experts. Based on the descriptive model, a system dynamics simulation model was built, incorporating existing approaches and developing eight interconnected modules. A sensitivity analysis attested that the model works according to existing literature and expert knowledge in production engineering. Lastly, we gathered data from an industrial ramp-up and confirmed the model's validity, with some limitations. To the best of our knowledge, this is the first openly available system dynamics model to model ramp-up behaviour that is validated through expert interviews and industry data.

However, the chosen control factors are limited to four and thus a manager's action space is small, although the event space is large due to the model's number of disturbance factors. Future research could thus implement more action options into the model. However, in system dynamics models it is key to attain a good level of complexity: too low and the results are trivial, too high and variables' causes and effects are incomprehensible. We consider the presented model as being on the verge of becoming too complex, as we have highlighted in the result section. Therefore, if future research implements more control factors, it should be reviewed if some variables can be omitted to reduce complexity.

We have provided production data from an industry ramp-up case. We conclude that the system dynamics model is capable of modelling realistic ramp-up behaviour. However, different ramp-up types exist that might require bespoke models, while our model is based on a generic case. Additionally, most model variable values could not be obtained from the industry partner. Future research should investigate not only "what is required" from a theoretical perspective, but also "what is possible" from a practical perspective. Therefore, a data model is needed that defines what data can be provided from a shop floor that is relevant to ramp-up performance prognosis.

One future research opportunity to realize said data model is to develop a digital shadow for ramp-ups. Current digital twins often represent a static production system, but during ramp-up, the production system undergoes frequent changes. Moreover, the system dynamics model can produce synthetic data that can be used to train machine learning algorithms. When the model is connected to shop floor data via a digital twin, either the system dynamics model or trained algorithm could be employed to make predictions about the course of the production ramp-up and allow early corrective measures. Lastly, we deem that there is no single source of truth regarding models. Future research could explore different approaches, such as discrete event simulation, for a more detailed operational perspective and combine findings with our system dynamics strategy perspective to add more valuable insights.

**Supplementary Materials:** The following supporting information can be downloaded at: https://www.mdpi.com/article/10.3390/systems12120575/s1, Text S1: SD Model Equations.

**Author Contributions:** Conceptualization, J.H.; methodology, J.H.; software, J.H. and B.K.; validation, J.H.; formal analysis, J.H.; investigation, J.H. and B.K.; resources, J.H., A.G. and R.H.S.; data curation, J.H. and B.K.; writing—original draft preparation, J.H.; writing—review and editing, A.G.; visualization, J.H. and B.K.; supervision, A.G.; project administration, J.H.; funding acquisition, R.H.S. All authors have read and agreed to the published version of the manuscript.

**Funding:** This research was funded by the Deutsche Forschungsgemeinschaft (DFG, German Research Foundation)—471703091.

**Data Availability Statement:** Model equations are available in Supplementary Material S1. Furthermore, the model is available upon request as vensim.mdl file. The dataset of the industry use case cannot be provided due to confidentiality restrictions.

**Conflicts of Interest:** The authors declare no conflicts of interest. The funders had no role in the design of the study; in the collection, analyses, or interpretation of data; in the writing of the manuscript; or in the decision to publish the results.

## Appendix A

$$\text{Demand rate} = \frac{Production\ demand}{1 + a * e^{-bt}} \tag{A1}$$

With a and b as model parameters that determine the sigmoid curve shape. t represents time of the ramp-up.

$$\begin{aligned} \text{dicretized\_input\_variable} = \text{IF THEN ELSE (Time} < \text{FINAL TIME}/3, \\ \text{w1} * \text{input\_variable, IF THEN ELSE (Time} < \text{FINAL TIME}*2/3, \\ \text{w2} * \text{input\_variable, w3} * \text{input\_variable)}) \end{aligned} \tag{A2}$$

$$\begin{aligned} \text{discretized production personnel} = \text{IF THEN ELSE(Time} < \text{FINAL TIME}/3, \\ 0.75*\text{production personnel, IF THEN ELSE(Time} < \text{FINAL TIME}*2/3, \\ \text{production personnel, production personnel))}. \end{aligned} \tag{A3}$$

**Table A1.** Overview of control factor changes for sensitivity analysis. The base run reflects model execution on default settings, while low and high runs reflect lower and higher settings, respectively. Each run is represented by three settings, one for each phase of the ramp-up.

| Control Factors | Low Run | Base Run | High Run |
|---|---|---|---|
| Avg. Training Time | 2–0.67–0.2 | 3–1–0.3 | 4–1.33–0.4 |
| Engineering Changes | 4–2–1 | 6–2–1 | 8–3–1 |
| Production Personnel | 14–18–18 | 15–20–20 | 16–22–22 |
| Maintenance Personnel | 4–4–8 | 5–5–10 | 6–6–12 |

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
