# Peer review of "A System Dynamics Stability Model for Discrete Production Ramp-Up"

_systems, doi:10.3390/systems12120575_

Round 1
Reviewer 1 Report
Comments and Suggestions for Authors
Nice paper, could improve a little with respect to:
1. Improve readability of figures with larger scriptsize for Fig. 4,5,6
2. The OEE calculation is not false, but there a better ways, based on a KPI network (e.g. there are multiple extensions and forecasting etc. based on an ISO standard)
3. Strong discussion but a bit more outlook, e.g. integration with digital twins to use it for foresight or adaptations, would be even better
Author Response
Comment 1: "Improve readability of figures with larger scriptsize for Fig. 4,5,6."
Response 1: We increased font size of figures 4,5, and 6.
Comment 2: "The OEE calculation is not false, but there a better ways, based on a KPI network (e.g. there are multiple extensions and forecasting etc. based on an ISO standard)"
Response 2: We based ourselves on the definition from TPM (Nakajima, 1988), but it is a good point and we might add it to the discussion!
Comment 3: "Strong discussion but a bit more outlook, e.g. integration with digital twins to use it for foresight or adaptations, would be even better"
Reponse 3: We have extended the paragraph of the discussion that elaborates more on the outlook and possibilities, especially regarding the use of digital twins , p. 16, l. 532ff
Reviewer 2 Report
Comments and Suggestions for Authors
This is a really interesting paper that advances our understanding and our ability to further our understanding of ramp-up. Given its prevalence in industry there is a corresponding absence of work in academia. This is an important contribution to an under represented area of research.
The paper is thorough in its treatment on the challenge, the available literature and the developed model. The work has a rigour throughout. In general.
There are a few areas that the authors could improve on. These would enhance the quality of the work with not much effort. For that reason, minor corrections have been recommended.
The selection of system dynamics is a valid one and the rationale for its fit is given. Why not, however, other techniques? A short piece on why not (say) discrete event simulation (DES) would be useful. To play the devil's advocate, DES is suitable for discrete production systems modelling, can accept the inputs (or adaptations of them) of those cited in the paper and importantly the discrete nature of the technique allows the step changes such as engineering change to be introduced and the cause and effect traced. This is not to criticise the choice of SD, it's to criticise the lack of justification of SD.
Second is the choice of parameters. They are justified through the use of the literature which is to be commended. In part they are significantly based on prior work which is to be commended. But then there is an industrial engagement to inform the choices. I think that text can be moved up to strengthen the choice before the models are created because, if I understood the informants, they were providing inputs and some relationships, they were not validating the modules you described prior. Or at least that is how I interpreted this really important paragraph immediately after figure 14.
Third is the module development. For existing modules, were they modified? If so how and why? For new modules, what was the development process? Objective? Logical? Was there testing or rigour in the validation of the modules? For example was there expert validation? You say later that it was a real production system (or imply that). If your answer to the development of the modules was based on working in the production system and iteratively developing the system then perhaps you should be open at say this (I'm sure I didn't miss this)
Finally, the complexity module seemed strange to me as I wasn't convinced it was really a module but an input or initialisation module. To me the complexity is embedded in the other modules such as worker skill. Perhaps I misunderstood the point.
Overall there is a feeling for me that something really complex has been created but the first test was as a whole SD modelling tool and therefore there modelling tool itself is complex and could create unusual or unexplainable behaviour. Perhaps if my points above are address this point is no longer valid.
I think all the above points can be explained quickly and succinctly. I see my criticisms are of an explanatory nature and not a fundamental research nature.
I wish you well in the rest of the review process and hope to be able to read and share your work soon! Thank you.
Author Response
Comment 1: "The selection of system dynamics is a valid one and the rationale for its fit is given. Why not, however, other techniques? A short piece on why not (say) discrete event simulation (DES) would be useful. To play the devil's advocate, DES is suitable for discrete production systems modelling, can accept the inputs (or adaptations of them) of those cited in the paper and importantly the discrete nature of the technique allows the step changes such as engineering change to be introduced and the cause and effect traced. This is not to criticise the choice of SD, it's to criticise the lack of justification of SD."
Response 1: You're completely right. In a publication that is currently in press [38] we compared various modeling methods in terms of suitability to our case. But failed to elaborate it properly in this work. We chose system dynamics as it is especially designed for complex systems and can model "fuzzy" factors, such as a worker skill. From our experience, DES is predominantly used for more concise cases, e.g., on the shop floor where things can be quantified. I have added a paragraph in "materials and methods" to account for that reasoning. And in fact, we wrote a research proposal for a hybrid modelling approach, that combines the benefits of both SD and DES and now added it to the discussion/outlook. We have also added a paragraph from p. 4, l. 169ff for explaination.
Comment 2: "Second is the choice of parameters. They are justified through the use of the literature which is to be commended. In part they are significantly based on prior work which is to be commended. But then there is an industrial engagement to inform the choices. I think that text can be moved up to strengthen the choice before the models are created because, if I understood the informants, they were providing inputs and some relationships, they were not validating the modules you described prior. Or at least that is how I interpreted this really important paragraph immediately after figure 14."
Response 2: Yes, we agree that the paragraph can be moved into the methodology section to underline our parameter choices. We located in the results section, since it is a result of a method (semi-structured expert interviews), but we would ultimately leave the placement decision to the editor, what fits best for the journal.
Comment 3: "Third is the module development. For existing modules, were they modified? If so how and why? For new modules, what was the development process? Objective? Logical? Was there testing or rigour in the validation of the modules? For example was there expert validation? You say later that it was a real production system (or imply that). If your answer to the development of the modules was based on working in the production system and iteratively developing the system then perhaps you should be open at say this (I'm sure I didn't miss this)"
Response 3: The existing modules were integrated based on their logic (e.g., A affects B positively or negatively) but were extended if deemed necessary to interact with the newly developed models. The validation with real production system data only occured at the end, as a final validation so to speak. We have added a paragraph for explaination on p. 5, l. 207 ff.
Comment 4: "Finally, the complexity module seemed strange to me as I wasn't convinced it was really a module but an input or initialisation module. To me the complexity is embedded in the other modules such as worker skill. Perhaps I misunderstood the point."
Response 4: You are right, the nomenclature is a little bit misleading. The overall SD model is designed to account for complexity of the whole ramp-up. The complexity "module" is named after Gartzen, who presentend a high level framework (morphological box) of what factors cause complexity in a ramp-up. Thus, the complexity module rather defines the starting conditions of the ramp-up and its initial complexity. Therefore, we renamed it to "structural complexity module" and elaborated more on its use from p. 11, l. 375ff.
Reviewer 3 Report
Comments and Suggestions for Authors
This paper proposes a system dynamics model that captures the unique dynamics of ramp-up phases by integrating stability and disturbance factors that influence key performance indicators overall equipment effectiveness, process capability, and production output. The system dynamic simulation results indicate that control factors realistically influence production system behavior during different ramp-up phases.
A major concern is the novelty of this paper. Since the major contribution is developing a simulation model, the possible novelty lies in either the new modeling/verification/… techniques and innovative algorithms perspective or practical application perspective that should provide new insights or solutions for the industry challenges. This paper has not clearly specified the novelty part yet.
Other small issues:
(1) In Section 2, pls add a paragraph to indicate the innovation of this paper; Plus, I’m not sure this type of literature review (referring to another review paper by the same author, as “An extract of the results is provided in this section.”) is appropriate for an independent paper;
(2) pls mention the tools used for developing system dynamics models somewhere in the paper;
(3) The discussion part focused on the shortcomings, but since this is also the conclusion part, it’s better to give a summary of the contributions. The first paragraph in this section is not adequate.
Author Response
Comment 1: "
A major concern is the novelty of this paper. Since the major contribution is developing a simulation model, the possible novelty lies in either the new modeling/verification/… techniques and innovative algorithms perspective or practical application perspective that should provide new insights or solutions for the industry challenges. This paper has not clearly specified the novelty part yet. Other small issues: (1) In Section 2, pls add a paragraph to indicate the innovation of this paper; Plus, I’m not sure this type of literature review (referring to another review paper by the same author, as “An extract of the results is provided in this section.”) is appropriate for an independent paper"Reponse 1: We have added sentences/paragraphs to highlight the novelty (p. 3, l. 120ff and p. 15, l. 504 ff ). Especially focusing, that our model tackles the issue that cited work focused on narrow aspects of ramp-up and not overall complexity. Further, our model is the only one to the best of our knowledge that is made openly available and is validated by expert interviews and industry data. The passage you quoted was probably misleading; we agree and thus we have removed it. The systematic literature review of this paper is unique work and not a self-cite of a previous systematic review (that we also cited). The sentence was meant to underline, that not all results of the systematic review are shown, for example the review process according to PRISMA, descriptive results etc. because the focus of this paper is the system dynamics model, not the literature review.
Comment 2: "(2) pls mention the tools used for developing system dynamics models somewhere in the paper;"
Response 2: You are completely right, we have added the used tools to the methodology section (p. 5, l. 199ff).
Comment 3: "The discussion part focused on the shortcomings, but since this is also the conclusion part, it’s better to give a summary of the contributions. The first paragraph in this section is not adequate."
Response 3: "Per the journals guidelines, a conclusion is optional, but not necessary. We therefore opted for just a discussion section and include the shortcomings there. But you are correct, thank you! We have focused more on the contributions and rewrote some aspects of the discussion (p.15, l. 504ff). If the editor suggests to add a conclusion instead, we might also do that. "
Reviewer 4 Report
Comments and Suggestions for Authors
The article is interesting and presents originality in its contribution to academia and society.
I do some recommendations for adjustments to improve the robustness of the research:
1) The objective stated in the abstract should be into the Introduction section of the article. The abstract is a complete part of the article and not part of the content developed in the research. Therefore, the abstract should include parts of the text of the article and not additional text. This is the identified objective "We propose a system dynamics model that captures the unique dynamics of the acceleration phases, integrating stability and disturbance factors that influence the main indicators of formation, overall equipment effectiveness, process capacity and production."
2) The concept of cause and effect applied in simulation with system dynamics seeks to establish a relationship with as many variables as possible, preferably establishing the connection relationship between them. Examples: Figure 10 shows that the Production Rate variable is not related to the Production Capacity variable. In figure 13, the same construction is seen in the relationship between the Run Time Losses, Production Rate and Short Downtime variables, considering that machine downtime influences the production rate, these variables need to be connected by the operational performance relationship.
Author Response
Comment 1: "The objective stated in the abstract should be into the Introduction section of the article. The abstract is a complete part of the article and not part of the content developed in the research. Therefore, the abstract should include parts of the text of the article and not additional text. This is the identified objective "We propose a system dynamics model that captures the unique dynamics of the acceleration phases, integrating stability and disturbance factors that influence the main indicators of formation, overall equipment effectiveness, process capacity and production."
Response 1: To the best of our knowledge, a good abstract contains significance, status quo, gap, contribution/aim, method(s), result(s) and implication/outlook. The aforementioned sentence is our formulation of aim/contribution, showing that out model addresses the gap. We agree that the abstract should not contain new information. But we consider our sentence from p.2, l. 63 ff as a formulation of aim/objective in the text. If that is not clear enough, we would consider rewriting it.
Comment 2: "The concept of cause and effect applied in simulation with system dynamics seeks to establish a relationship with as many variables as possible, preferably establishing the connection relationship between them. Examples: Figure 10 shows that the Production Rate variable is not related to the Production Capacity variable. In figure 13, the same construction is seen in the relationship between the Run Time Losses, Production Rate and Short Downtime variables, considering that machine downtime influences the production rate, these variables need to be connected by the operational performance relationship"
Response 2: We agree only partly. System Dynamics seeks to explain or model a system with just the right grade of complexity. Too little complexity and one does not get enough valuable insight from the system, too high complexity and the understanding of the system gets blurry, as cause and effects cannot be attributed anymore. Thus, seeking as many connections as possible might lead to the second scenario in our opinion.
In your given example, Production Capacity is not directly connected to the production rate, but influences the Potential Production Performance, which in turn directly influences Production Rate. Therefore, we deem that the effect of "Production Capacity" is sufficiently reflected in the Production Rate. As with the second example, "Short downtimes" directly influence "Production Capacity" and are thus also affecting the production rate. Of course, relationships can be modified, the result is a different model however.
Reviewer 5 Report
Comments and Suggestions for Authors
According to the theme of this work, authors may consider a bit on model prediction also. Conditioning with previous data, some related factors with weights parameters and so on may relate from the practical point of view.
In general, some step-size parameters may play significant role of its performance. In this work, how this relate such that the setting of Adaptation rate =0.003 in the supplement material!
Authors did some literature survey. Nonetheless, in order to emphasize the impact on current research, some recently references within 2024 may help.
One of importance players is the demand rate which may significantly impact. Therefore, authors should consider to demonstrate this point within the validation results (if possible).
In the third paragraph of the discussion, is it the limitation of the proposed scheme?
Author Response
Comment 1: "According to the theme of this work, authors may consider a bit on model prediction also. Conditioning with previous data, some related factors with weights parameters and so on may relate from the practical point of view. "
Response 1: Thank you for pointing that out, we have extended our discussion to account more for possible model prediction in the future!
Comment 2: "In general, some step-size parameters may play significant role of its performance. In this work, how this relate such that the setting of Adaptation rate =0.003 in the supplement material!"
Response 2: We agree that the design of some parameters is contestable. One difficulty of this work was to quantify intangible factors, for example how fast workers learn new ways of manufacturing, i.e., adapt to new circumstances. Without empirical or theoretical models, some procedures had to be designed to our best knowledge and estimates. We have added that to our critical discussion section. But it remains an opportunity for future models to replace our modules with more sophisticated ones.
Comment 3: "Authors did some literature survey. Nonetheless, in order to emphasize the impact on current research, some recently references within 2024 may help. "
Response 3: You are correct. Taking also another reviewers comment into account, we have added a paragraph in section 2 to highlight novelty and impact of this paper (p. 3, l. 120 ff). However, there are no similar modeling approaches that were published within the last couple of years to the best of our knowledge.
Comment 4: " One of importance players is the demand rate which may significantly impact. Therefore, authors should consider to demonstrate this point within the validation results (if possible)."
Response 4: Yes, the demand rate determines how much production is planned. So we modeled it to the best of our knowledge from literature and interviewing industry experts. Unfortunately, we could not obtain real order data from the industry. We are thankful for clues for any open data source for that matter.
Comment 5: "In the third paragraph of the discussion, is it the limitation of the proposed scheme?"
Reponse 5: It is a limitation for a potential deployment in an industry environment, yes. We have extended our "discussion" section ( p. 16, l. 532 ff) to go more into detail about future research opportunities and underlining the need for a ramp-up data model approach.
Round 2
Reviewer 3 Report
Comments and Suggestions for Authors
Please use numbering such as 2.1, 2.2, 3.1, 3.2, etc., to structure the subsections more clearly. I have no other questions.
Author Response
Comment 1: Please use numbering such as 2.1, 2.2, 3.1, 3.2, etc., to structure the subsections more clearly. I have no other questions.
Answer 1: Thank you for pointing that out! We adjusted the section numbering to fit the journal guidelines. Maybe that could be incorporated into the provided journal template in the future.
Reviewer 5 Report
Comments and Suggestions for Authors
Authors already considered all comments raised by the reviewer.
Only some points which may be required by the editor office . if it's necessary.
Author Response
Comment 1: Authors already considered all comments raised by the reviewer. Only some points which may be required by the editor office . if it's necessary.
Answer 1: Thank you! We look forward to the editor's comments.